# Double-Exposure Method with Synchrotron White X-ray for Stress Evaluation of Coarse-Grain Materials

**Kenji Suzuki** [1,*] **, Ayumi Shiro** [2] **, Hidenori Toyokawa** [3] **, Choji Saji** [3] **and Takahisa Shobu** [4]

1   Faculty of Education, Niigata University, Niigata 950-2181, Japan
2   National Institutes for Quantum and Radiological Science and Technology, Sayo-gun,
    Hyogo 679-5148, Japan; shiro.ayumi@qst.go.jp
3   Japan Synchrotron Radiation Research Institute, Sayo-gun, Hyogo 679-5198, Japan;
    toyokawa@spring8.or.jp (H.T.); saji@spring8.or.jp (C.S.)
4   Japan Atomic Energy Agency, Sayo-gun, Hyogo 679-5148, Japan; shobu.takahisa@jaea.go.jp
*   Correspondence: suzuki@ed.niigata-u.ac.jp

**Abstract:** Stress measurements of coarse-grained material are difficult using synchrotron X-ray diffraction because the diffraction patterns of coarse-grained materials are spotty. In addition, the center of the diffraction pattern is unknown for the transmitted X-ray beam. Here, a double-exposure method is proposed as the countermeasure against this issue. In the experiment, we introduce a CdTe pixel detector. The detector is a newly developed area detector and can resolve high-energy X-rays. The strains of the coarse-grained material can be measured using a combination of the double-exposure method, white synchrotron X-ray, and the CdTe pixel detector. The bending stress in an austenitic stainless steel plate was measured using the proposed technique. As a result, the measured stress corresponded to the applied bending stress.

**Keywords:** synchrotron white X-ray; double-exposure method; CdTe pixel detector; coarse grain; X-ray stress measurement; austenitic stainless steel

## 1. Introduction

A high-energy synchrotron X-ray between 30 keV and 1 MeV has a large penetration depth and directivity, and therefore has an excellent beam quality [1]. Internal stresses can be measured using strain scanning with hard synchrotron X-rays, and the method is used widely in industry [2]. However, the use of the strain scanning method is limited as the diffraction ring from the gauge volume should be continuous.

The diffraction pattern from coarse grains is spotty, and an area detector must be used to measure the diffraction spots. In addition, a special slit system is needed to create the gauge volume for the area detector. Some 2D slit systems have been developed, e.g., the conical slit and the spiral slit [3–6]. The stress measurement technique of a 2D detector with a 2D slit system is required for evaluating the internal stresses of textured materials or those with coarse grains. The authors also proposed a diffraction spot trace method (DSTM) as the countermeasure [7,8]. In the DSTM, scanning the gauge-volume that is made by the rotating slit-system in the sample, the diffraction spots using an area detector are recorded. Both of the intensity and position of each diffraction spot have to be traced. The DSTM is useful for measuring stresses in coarse-grained materials or welded metals. However, the rotating slit-system of the DSTM is complicated, and the analysis for the DSTM is difficult.

Therefore, we propose a double-exposure method (DEM). The DEM does not require a complicated slit system, but it does require an area detector. The distribution of the residual stress in coarse-grained material can be measured using the DEM, and the effectiveness of the DEM

was examined in our previous study [9], where the DEM was used to measure the stresses of the coarse-grained materials. The DEM is a creative technique, but the X-ray energy is limited to 30 keV as it uses a Si sensor pixel detector (PILATUS detector [10]).

On the other hand, three-dimensional X-ray diffraction (3DXRD) is novel technique and a comprehensive account of the 3DXRD has been summarized [11]. The DEM does not require a rotation of a sample as compared with the 3DXRD.

Techniques of Laue micro X-ray diffraction have developed for about 15 years. Software suites for Laue pattern analysis are available, e.g., XMAS [12] and Laue Tools [13]. Strain analysis has been studied using these techniques [14–16].

In this study, we propose a stress measurement using a CdTe pixel detector. The CdTe pixel detector can recognize high-energy X-rays. Therefore, a diffraction pattern like monochromatic X-rays can be obtained from a Laue pattern using the CdTe pixel detector. The stresses of coarse-grained materials were measured using a combination of the DEM and the CdTe pixel detector with high-energy white X-rays.

## 2. Methods

### 2.1. Double-Exposure Method

When we use a diffractometer or an area detector, the unique diffraction center is assumed. However, this assumption is adopted in the case of materials such as powders, and depends on a mean effect by a large number of grains. The diffraction ring from powders is shown as a continuous ring. In coarse-grained materials, it is difficult to determine the diffraction centre because the number of grains is limited. The diffraction from coarse-grained materials shows a spotty pattern. Therefore, we have to measure the strain of the grain using the diffraction spot.

We consider that an X-ray beam transmits the sample as shown in Figure 1a. The line $l_X$ is the X-ray beam, and the line $l_D$ is the beam diffracted by one grain in the specimen. The diffraction spot from the grain is detected at two positions, $P_1$ and $P_2$, using an area detector (CdTe pixel detector). If we know the two positions at $P_1(x_1, y_1, z_1)$ and $P_2(x_2, y_2, z_2)$, the line $l_D$ can be determined. The angle formed between the line $l_X$ and $l_D$ is the diffraction angle $2\theta$.

The diffraction radius at $P_1$, $r_1$ and the diffraction radius at $P_2$, $r_2$ are given by the following equations:

$$r_1 = \sqrt{x_1^2 + y_1^2}, \qquad r_2 = \sqrt{x_2^2 + y_2^2}. \tag{1}$$

The distance between $P_1$ and $P_2$ is $L$ $(= z_2 - z_1)$, so the diffraction angle $2\theta$ can be obtained by the following:

$$2\theta = \arctan\left(\frac{r_2 - r_1}{L}\right). \tag{2}$$

Figure 1b shows details of the intersection between the lines $l_X$ and $l_D$. The unit vectors of the lines $l_X$ and $l_D$ are expressed by $e_X$ and $e$, respectively, where each unit vector is given by the following:

$$e_X = (0, 0, 1), \tag{3}$$

$$e = (x_2 - x_1, y_2 - y_1, z_2 - z_1)\frac{\cos(2\theta)}{L}. \tag{4}$$

As shown in Figure 1b, the vectors $\overrightarrow{OO_X}$ and $\overrightarrow{OP_C}$ are:

$$\overrightarrow{OO_X} = -l_0\, e_X, \tag{5}$$

$$\overrightarrow{OP_C} = \overrightarrow{OP_1} - l_1\, e. \tag{6}$$

Therefore, the vector $\overrightarrow{P_C O_X}$ is obtained from the equation:

$$\overrightarrow{P_C O_X} = \overrightarrow{OO_X} - \overrightarrow{OP_C} = -l_0\, e_X - \overrightarrow{OP_1} + l_1\, e . \tag{7}$$

Since the vector $\overrightarrow{P_C O_X}$ is normal to the line $l_D$, the inner product between the vectors $\overrightarrow{P_C O_X}$ and *e* becomes zero.

$$e \cdot \overrightarrow{P_C O_X} = 0 . \tag{8}$$

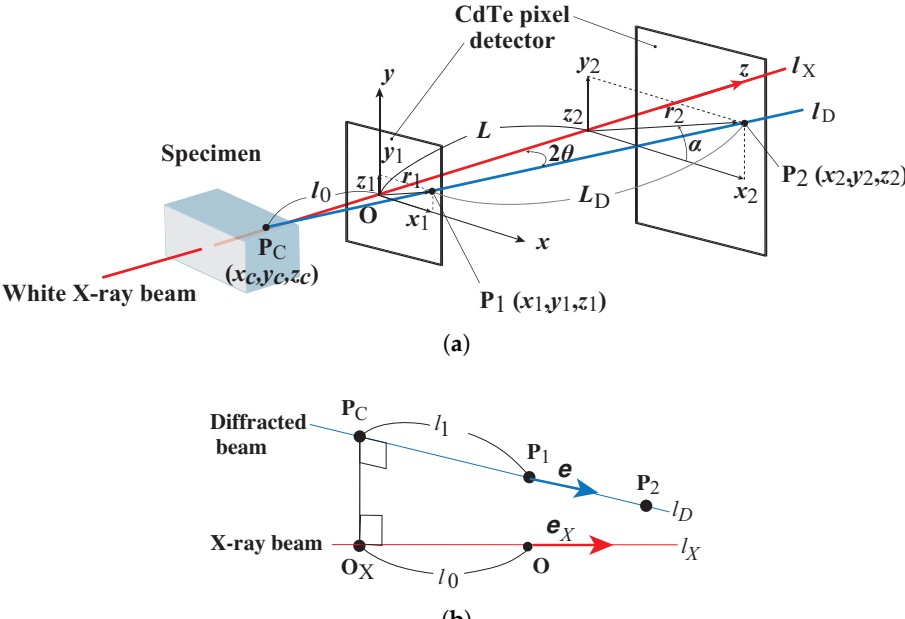

**Figure 1.** Double-exposure method with a synchrotron white X-ray source. (**a**) The line $l_X$ is the transmitted white X-ray beam and the line $l_D$ is the diffracted X-ray beam from a coarse grain. The diffracted beam is detected by the CdTe pixel detector at $P_1$ and $P_2$; (**b**) details of $P_C$, where the lines of $l_X$ and $l_D$ intersect. The line $\overline{P_C O_X}$ is normal to lines $l_X$ and $l_D$.

Equation (8) provides the relationship between $l_1$ and $l_0$:

$$l_1 = \frac{(x_2 - x_1)\, x_1 + (y_2 - y_1)\, y_1 + (z_2 - z_1)\, z_1}{L_D} + \frac{z_2 - z_1}{L_D}\, l_0 . \tag{9}$$

The line $l_X$ is also normal to the vector $\overrightarrow{P_C O_X}$ and the inner product between $l_X$ and $\overrightarrow{P_C O_X}$ becomes zero:

$$e_X \cdot \overrightarrow{P_C O_X} = 0 . \tag{10}$$

We obtain the following equation from Equation (10):

$$l_0 = \frac{z_2 - z_1}{L_D}\, l_1 - z_1 , \tag{11}$$

where the following relationships are given in Figure 1:

$$L_D = \frac{L}{\cos(2\theta)} , \quad L = z_2 - z_1 \text{ and } z_1 = 0 . \tag{12}$$

Solving the simultaneous Equations (9) and (11), we have the following equations about the unknowns, $l_0$ and $l_1$:

$$l_0 = \frac{\cot^2(2\theta)}{L} \left[ (x_2 - x_1)\, x_1 + (y_2 - y_1)\, y_1 \right],$$ (13)

$$l_1 = \frac{\cos(2\theta)}{L \sin^2(2\theta)} \left[ (x_2 - x_1)\, x_1 + (y_2 - y_1)\, y_1 \right].$$ (14)

The intersection $P_C(x_C, y_C, z_C)$ can be led using $l_0$ and $l_1$.

$$\begin{pmatrix} x_C \\ y_C \\ z_C \end{pmatrix} \overrightarrow{OP_C} = \overrightarrow{OP_1} - l_1\, e = \begin{pmatrix} x_1 \\ y_1 \\ z_1 \end{pmatrix} - l_1 \frac{\cos(2\theta)}{L} \begin{pmatrix} x_2 - x_1 \\ y_2 - y_1 \\ z_2 - z_1 \end{pmatrix}$$ (15)

Finally, we obtain the intersection $P_C(x_C, y_C, z_C)$:

$$x_C = x_1 + \frac{z_C}{L} (x_2 - x_1)$$ (16)

$$y_C = y_1 + \frac{z_C}{L} (y_2 - y_1)$$ (17)

$$z_C = - \frac{\cot^2(2\theta)}{L} \left[ (x_2 - x_1)\, x_1 + (y_2 - y_1)\, y_1 \right].$$ (18)

We notice that the intersection $P_C$ gives us the diffraction position for each grain.
In addition, the azimuth angle of the diffraction beam $\alpha$ is given by the following:

$$\alpha = \arctan \left( \frac{y_1}{x_1} \right).$$ (19)

As mentioned above, we obtain the diffraction angle $2\theta$ and the intersect position $P_C(x_C, y_C, z_C)$ using the DEM.

### 2.2. CdTe Pixel Detector

We developed the CdTe pixel detector to measure the energy-resolved diffraction image using synchrotron white X-rays. The CdTe pixel detector is shown in Figure 2.

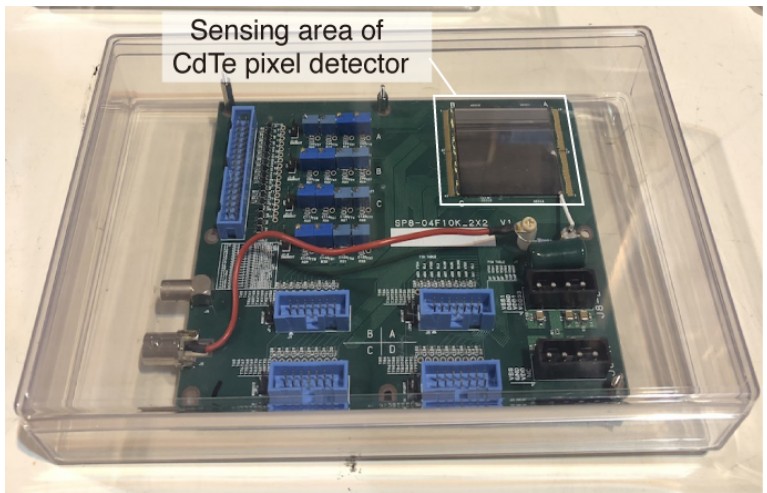

**Figure 2.** CdTe pixel detector. The sensing area is on the upper-right side of photograph.

The CdTe sensor has the Pt/CdTe/Al configuration using a high-resistivity p-type single-crystal CdTe wafer that is 750 μm thick. The front side (X-ray irradiated side) was processed with a single platinum electrode. The back side was deposited with a $95 \times 100$ matrix of aluminum electrodes.

The pixel format of the photon-counting application specific integrated circuit (ASIC), SP8-04F10K, is $95 \times 100$. The CdTe pixel detector is combined with an aluminum Schottky diode sensor and the photon-counting ASIC [17,18].

In this study, a new CdTe pixel detector was prepared as shown in Figure 2. It is composed of a $2 \times 2$ configuration of the SP8-04F10K hybrid detector by Sn-Bi solder bump bonding. Thus, the CdTe pixel detector has a wider sensor area than conventional detectors. The pixel dimensions are $0.2 \times 0.2$ mm$^2$, and the detected area is $40.2 \times 39.2$ mm$^2$ ($201 \times 191$ pixels). Each pixel of the detector has a 24-bit counter.

### 2.3. Experiments with Synchrotron White X-rays

The experiments using synchrotron white X-rays were conducted on beamline BL14B1 at SPring-8 in Hyogo, Japan. The CdTe pixel detector was mounted on the *xyz*-stages as shown in Figure 3. The dimensions of the incident white X-ray beam were $0.1 \times 0.1$ mm$^2$.

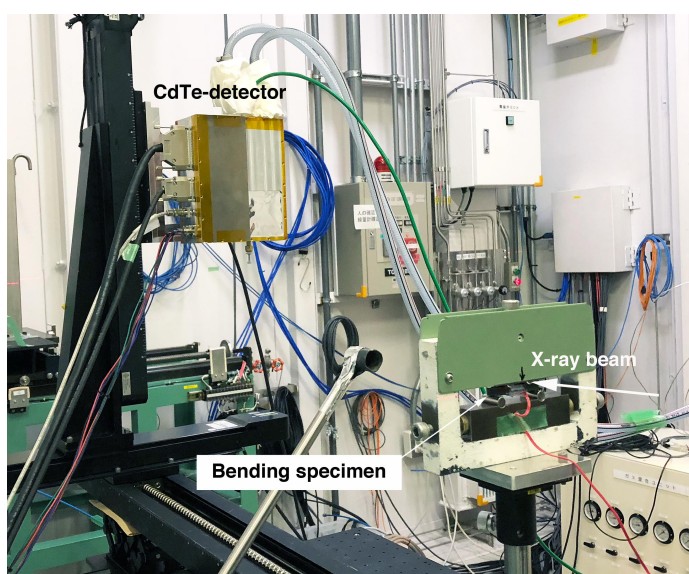

**Figure 3.** Experiment on beamline BL14B1 at SPring-8, Hyogo, Japan. The synchrotron white X-ray beam transmits the 4-point-bending specimen from the right side. The diffraction patterns are measured at locations $P_1$ and $P_2$ by moving the CdTe pixel detector through the stages.

As the area of the detector is small, the measured area of the diffraction image was formed by moving the detector. The diffraction image at position $P_1$ is composed of the $2 \times 3$ images of the detector, and position $P_2$ is composed of the $4 \times 5$ images of the detector. The length $L$ ($= z_2 - z_1$) is 500 mm.

The used specimen was a low-carbon forged austenitic stainless steel, SUSF316L, and its mean grain size was 300 μm. The dimensions of the specimen were a length of 45 mm, a height of 6 mm, and a width of 7.7 mm. To examine the DEM with synchrotron white X-rays, 4-point bending was applied to the specimen. The applied strain on the tensile surface (the lower side) was measured with a strain gauge. As shown in Figure 3, the synchrotron white X-ray beam transmits the 4-point-bending specimen from the right side. The strains from the upper side to lower side were measured with a step size of 0.5 mm.

## 3. Results and Discussion

### 3.1. Energy Calibration of CdTe Pixel Detector

The CdTe pixel detector is composed of four sensor segments of SP8-04F10K with the $95 \times 100$ pixels, as shown in Figure 4a. The gap between the segments was 0.2 mm. Each pixel

of the detector functions as an energy-resolved counter. A diffraction image using white X-rays includes a wide range of photon energies, and the pixels can count high-energy X-rays.

We considered that photons with the energy distribution as shown in Figure 4b enter the pixel counter. Since the pixel counter had one threshold-type comparator, only the photons with an energy higher than the threshold energy were counted. The threshold energy is controlled by the threshold voltage of the comparator as shown in Figure 4c. Increasing the threshold voltage, the threshold of energy becomes low. Photons with low energy can be counted as shown in the Figure 4c. As a result, the count of the pixel behaves as shown in Figure 4d. The count of the pixel with an increase in the threshold voltage is equivalent to the integrated value from the threshold X-ray energy of the X-ray diffraction to high energy. In this way, the CdTe detector can separate the photon energy and counts. The features of the CdTe pixel detector are the determination of the X-ray energy and detection of high-energy X-rays.

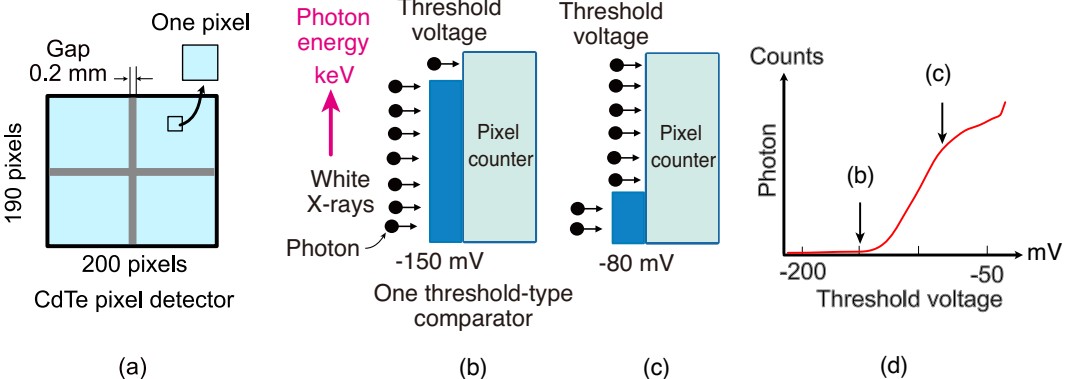

**Figure 4.** Mechanism of energy-resolved photon counting by the CdTe pixel detector. (**a**) CdTe pixel detector. Each pixel functions as an energy-resolved counter; (**b**) white X-rays include a wide energy range of wavelengths; (**c**) the energy of the counted photon is limited by the threshold voltage of one threshold-type comparator; (**d**) relationship between threshold voltage and counts of photons.

For the diffraction of SUSF316L, the dimensions of the white X-ray beam were $0.1 \times 0.1$ mm$^2$. The threshold voltages of the CdTe pixel detector ranged from $-210$ mV to $-70$ mV, with a step of 1 mV. The exposure time was 1 s for each step, and we measured 141 diffraction images. Figure 5 shows the Laue patterns measured by the CdTe pixel detector. Figure 5a, b were measured with threshold voltages of $-130$ mV and $-80$ mV, respectively. As the threshold voltage is lower, Figure 5a shows the Laue pattern with higher X-ray energies. The lower X-ray energies are also included in Figure 5b due to the higher threshold voltage. Therefore, the image measured by the CdTe pixel detector becomes brighter and the number of the diffraction spots increases with the increase in the threshold voltage.

The images in Figure 5 are raw images measured by the threshold voltage; therefore, we had to obtain the images using the threshold X-ray energy. To obtain the image using the threshold X-ray energy, the relationship between the threshold voltage and the threshold X-ray energy must be determined. This relationship is needed for each pixel. If the X-ray energy image is obtained, we can measure the strain from it. To obtain the relationship between the threshold voltage and the threshold X-ray energy, the characteristic X-rays of Pb-*Kα* and W-*Kα* were used. The white X-rays were irradiated onto a Pb-foil and a W-foil, and these fluorescent X-rays were measured by the CdTe pixel detector. The comparator threshold voltage of each pixel was calibrated to the corresponding X-ray energies with fluorescent *Kα* X-rays of W (58.87 keV) and Pb (74.25 keV).

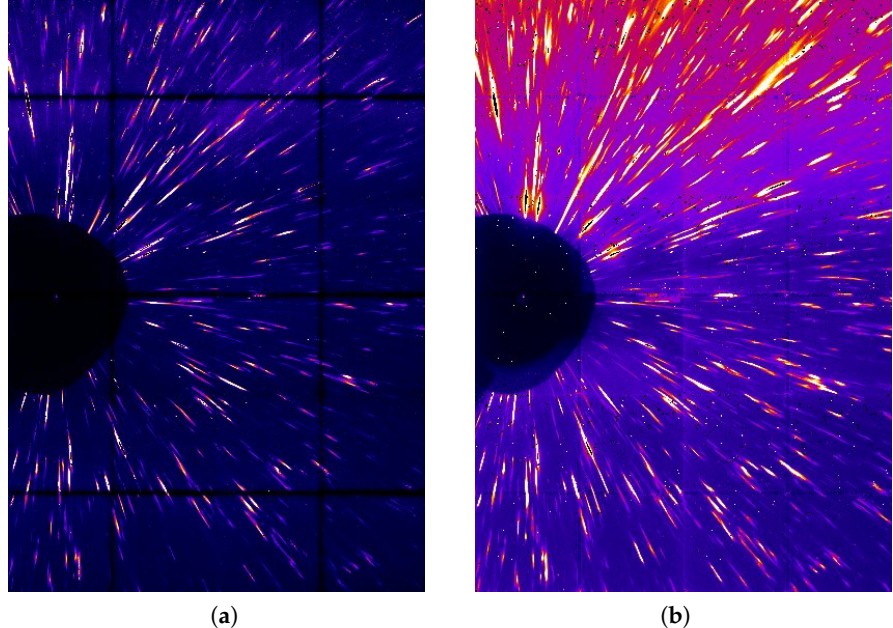

(**a**)            (**b**)

**Figure 5.** Laue patterns measured by the CdTe pixel detector at $P_1$. (**a**) Image measured with threshold voltage of $-130$ mV; (**b**) image measured with threshold voltage of $-80$ mV.

Figure 6 shows the relationship between the comparator threshold voltage and the X-ray intensity for the Pb-foil and W-foil. As shown in Figure 6a, the X-ray intensity measured by the detector increases near the threshold voltages of $-132$ and $-122$ mV due to the fluorescent X-rays of Pb-$K\beta$ and Pb-$K\alpha$, respectively. The different intensities from the threshold voltages indicate the peaks of the fluorescent X-ray of Pb-$K\beta$ and Pb-$K\alpha$. In addition, the peaks of the fluorescent X-rays of Ta and Sn are shown in the figure. Although the reason is not clear, it seems that the slits of Ta and Sn–Bi bump solder bonding of the detector caused these fluorescent X-rays.

Figure 6b demonstrates the changes in the X-ray intensity for pixel No. 3000 and its difference for the W-foil with the increase in the threshold voltage. The difference indicated by the red line shows the peaks of the fluorescent X-rays of W-$K\beta$ and W-$K\alpha$.

Since the peaks of Pb-$K\alpha$ and W-$K\alpha$ were strong and well defined, as shown in Figure 6, the X-ray energy of the CdTe pixel detector was calibrated using the peaks of Pb-$K\alpha$ and W-$K\alpha$. We assumed that the relationship between the threshold voltage $V_i$ and the threshold X-ray energy $E$ was a linear function for each pixel. Therefore, its approximation is expressed as follows:

$$E = a_i V_i + b_i, \quad (i = 1 \sim 38{,}000) \tag{20}$$

where $a_i$ and $b_i$ are coefficients determined from $V_i$ and $E$, respectively.

Figure 7 shows the results of the energy calibration of the CdTe pixel detector. The CdTe pixel detector was calibrated using Pb and W-$K\alpha$ characteristic X-rays before and after the experiments. The means and standard deviations of the threshold voltage for all pixels are shown in Figure 7a. The differences in the threshold voltage before and after the experiments were a few kilo-electron volts. We calibrated the X-ray energy of the CdTe pixel detector using the measured data obtained from the experiments.

For each pixel, the relationships between the X-ray energy and the threshold voltage were determined using the $K\alpha$ characteristic X-ray of Pb and W. As a result, the coefficients $a_i$ and $b_i$ in Equation (20) were obtained. Figure 7b shows the distributions of $a_i$ and $b_i$. Each pixel of the CdTe pixel detector was calibrated using $a_i$ and $b_i$ so that the threshold X-ray energy of the CdTe pixel detector, $E$ (keV), was determined using the threshold voltage $V_i$.

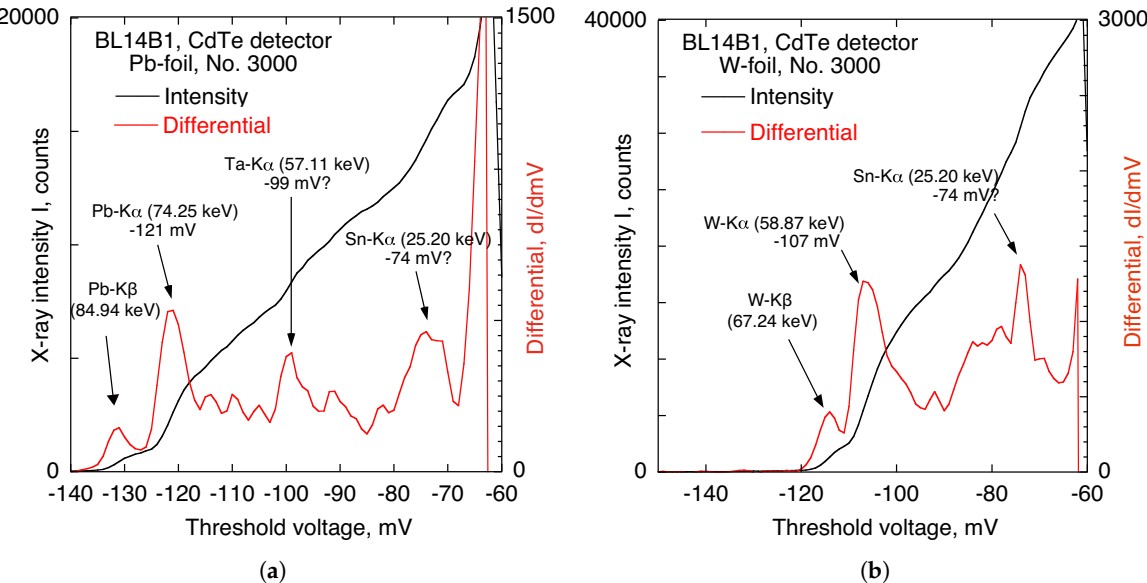

**Figure 6.** Relationship between the comparator threshold voltage and the X-ray energy. As an example, the result for the No. 3000 pixel is shown. The black line indicates the X-ray counts with the increase in the threshold voltage, and the red line indicates the difference using the threshold voltage. (**a**) Measured fluorescent X-rays for Pb-foil; (**b**) measured fluorescent X-rays for W-foil.

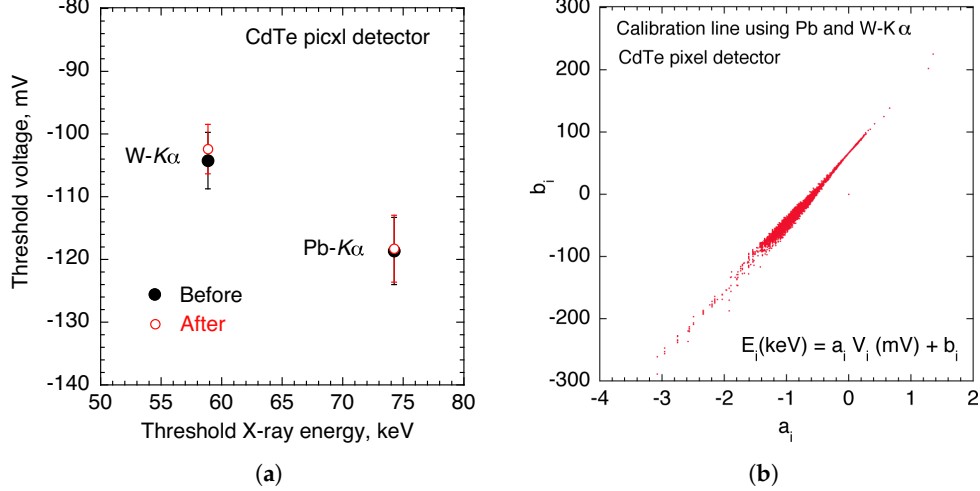

**Figure 7.** Results of the calibration of the CdTe pixel detector. (**a**) The means and standard deviations of threshold voltage for Pb and W-$K\alpha$ before and after the experiments; (**b**) distributions of coefficients $a_i$ and $b_i$, which indicate the relationship between the threshold X-ray energy and threshold value for each pixel. The coefficients $a_i$ and $b_i$ were determined using the characteristic X-ray energies of Pb and W-$K\alpha$.

### 3.2. Constructing Images by X-ray Energy

Images by the threshold X-ray energy are constructed from each pixel of the CdTe pixel detector. Therefore, a count of every pixel has to be prepared. The procedures for constructing images by the threshold X-ray energy $E$ are as follows:

1. A value of the threshold X-ray energy $E$ is determined.
2. The threshold voltage $V_i$ is determined using $E$, $a_i$, and $b_i$.
3. The X-ray intensity of the pixel is read out from the image measured with the threshold voltage $V_i$.
4. The above-mentioned operation is executed for all pixels.
5. This procedure is applied for every X-ray energy.

Figure 8 demonstrates these procedures. The numbers in the figure correspond to the above procedures.

The image of the position $P_2$ is composed of $4 \times 5$ scenes, and one scene is composed of $201 \times 191$ pixels. For example, the X-ray energy images from $-210$ mV to $-70$ mV are composed of a huge amount of data of $1.08 \times 10^8$. Constructing the images with the threshold X-ray energy from the image data with the threshold voltage is an important aspect of this work, and requires enormous computing times.

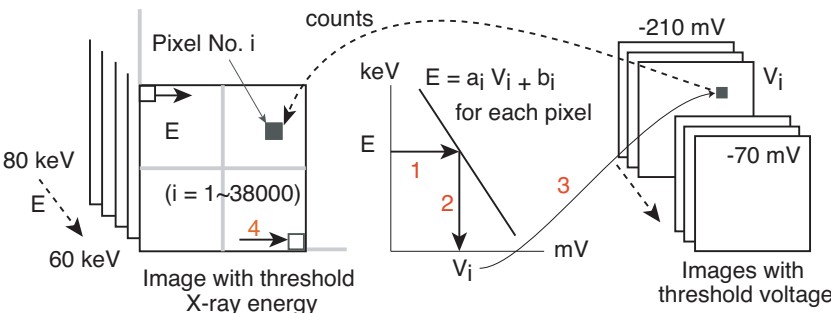

**Figure 8.** Constructing images by X-ray energy. The equation $E = a_i V_i + b_i$ is determined by the energy $E$ and the pixel No. $i$, and then the threshold voltage, $V_i$, is determined. The count at pixel No. $i$ is obtained from the image by threshold voltage $V_i$. These procedures are done for each pixel from No. 1 to No. 38,000.

The image with the threshold X-ray energy is depicted in Figure 9 as an example. Figure 9 is the Laue pattern of an austenitic stainless steel, and its image was constructed by white X-rays with a threshold value of 69 keV. Therefore, the image is equivalent to the Laue pattern generated by the white X-rays over 69 keV. The diffraction spot of the image with the threshold X-ray energy is not equal to an image with a monochromatic X-ray source. Upon decreasing the threshold X-ray energy, the diffraction spot shows a strong intensity and disappears with progression to a low diffraction angle. As a result, the diffraction pattern shows a radial shape. The diffraction spots on the image with the threshold X-ray energy behave in this manner.

To measure a lattice strain, a diffraction pattern like that obtained using monochromatic X-rays must be made using the threshold X-ray energy images. An X-ray intensity of $E_i$ (keV) is defined by $I(E_i)$. The intensities of the threshold X-ray energy $E_{i-1}$ and $E_{i+1}$ are defined by $I_{\text{th}}(E_{i-1})$ and $I_{\text{th}}(E_{i+1})$, respectively. The image of $I_{\text{th}}(E_i)$ is an integrated value from the threshold energy $E_i$ to a high energy. The image of $I(E_i)$ is calculated by the difference equation:

$$I(E_i) = \frac{I_{\text{th}}(E_{i-1}) - I_{\text{th}}(E_{i+1})}{2} . \tag{21}$$

The diffraction image with X-ray energy $E_i$ can be obtained by the difference between the threshold X-ray energy images of $I_{\text{th}}(E_{i-1})$ and $I_{\text{th}}(E_{i+1})$.

The diffraction images with X-ray energies from 60 to 80 keV were calculated by Equation (21). Figure 10a shows the diffraction image with an X-ray energy of 68 keV as an example. The image in Figure 10a is different from the image in Figure 9. The spots with a strong intensity in Figure 9 do not correspond to those in Figure 10a. For example, the diffraction spot at the lower left in Figure 9 has a strong intensity, but has a weaker intensity in Figure 10a. In the diffraction image for 68 keV in Figure 10a, the diffraction spots at the lower centre of the image show a strong intensity.

The outline of the diffraction spot was determined by 70% of the peak height, and the peak position was determined from the center of gravity of its outline. Its result is shown in Figure 10b. The peak position of each diffraction spot is indicated by a cross.

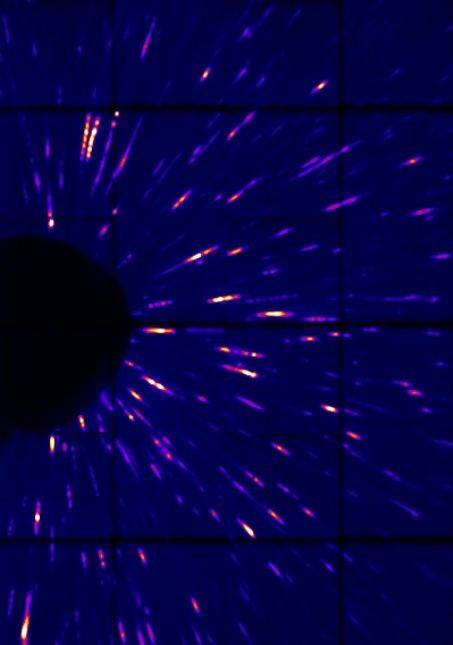

**Figure 9.** Laue pattern with threshold X-ray energy of 69 keV at position $P_1$. This diffraction image includes the diffraction points that were obtained over the threshold X-ray energy of 69 keV. The sample was an austenitic stainless steel.

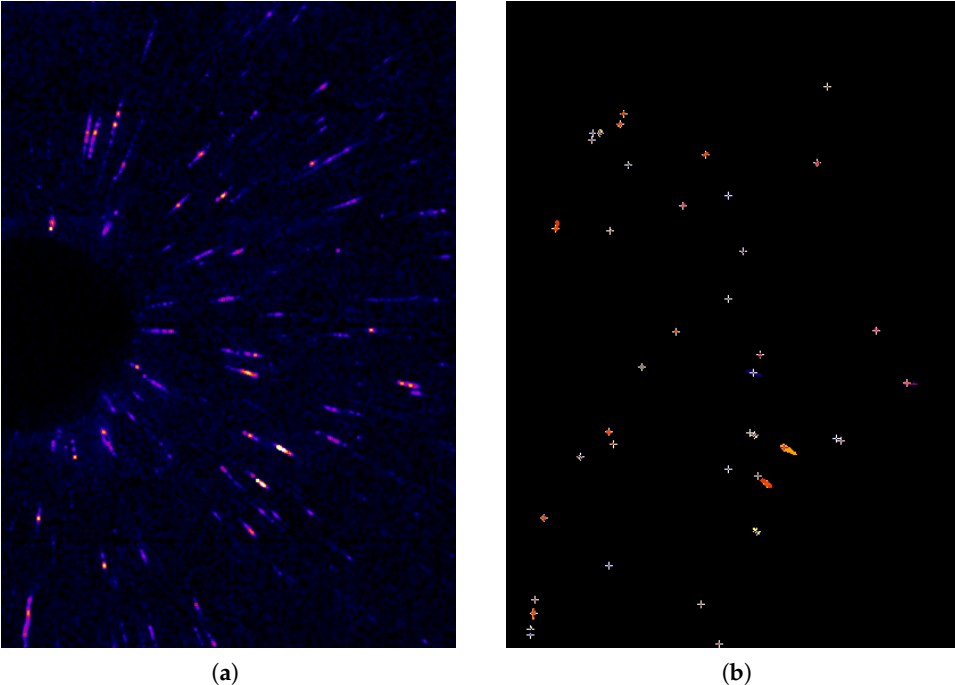

| (a) | (b) |

**Figure 10.** Diffraction image and peak positions with an X-ray energy of 68 keV. (**a**) Diffraction image calculated by the difference between threshold X-ray energy; (**b**) peak positions indicated by a cross.

However, the peak position and intensity of the diffraction spot changed with the change in the X-ray energy. The behavior of each diffraction spot of the X-ray energy images is demonstrated in Figure 11. The peak intensity of each diffraction spot shows the maximum, as shown in Figure 11. When the peak intensity is the maximum, the diffraction condition for the spot is the most suitable for Bragg's law. Therefore, the most suitable diffraction condition for each diffraction spot was determined by the X-ray wavelength at the maximum peak intensity.

In this study, we searched for the maximum of the peak intensity for each diffraction spot on $P_1$ and looked for the corresponding diffraction spot on $P_2$ for its X-ray wavelength. The diffraction angle $2\theta$ was determined from a pair of the diffraction spots on $P_1$ and $P_2$ using Equation (2).

When we searched for a pair of diffraction spots, we found some combinations. In addition, we found some combinations for the diffraction planes $h\,k\,l$. Therefore, unlike with the monochromatic X-ray, it was challenging to look for a pair of the diffraction spots. Eventually, we selected a pair of optimum spots according to the following:

1. The intersection distance is within 5 mm, i.e., $\overrightarrow{O_X\,P_C} = \sqrt{x_C^2 + y_C^2} \leq 5$.
2. The intersection position $z_C$ is within $-330$ to $-170$ mm.
3. The obtained lattice spacing $d$ has the consistency with the a lattice plane $h\,k\,l$ for a face-centered cubic system.

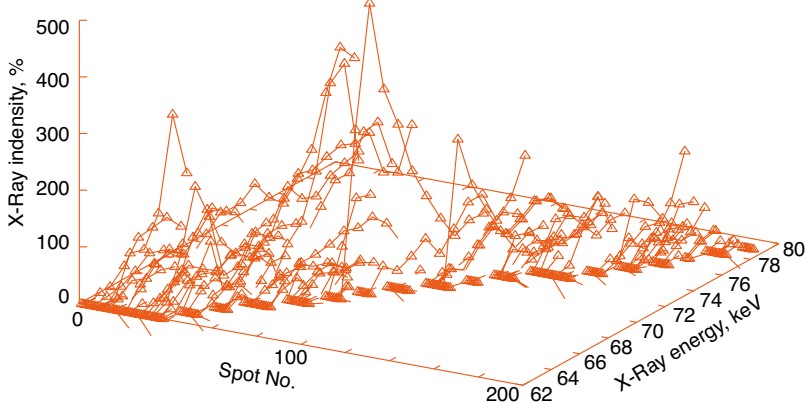

**Figure 11.** The intensity of each diffraction spot changes with the change in the X-ray energy. When the intensity is the maximum, the diffraction condition for the spot is the most suitable for Bragg's law.

### 3.3. Measurement of Bending Strains

We measured the bending strains of the rectangular bar. The diffraction angle $2\theta$ was calculated from the pair of diffraction spots obtained by the DEM with white X-rays and then the lattice spacing $d_{hkl}$ of the lattice plane $hkl$ was calculated. As a result that lattice spacing depends on a lattice plane, it is impossible to compare the lattice spacings for the different lattice planes. Therefore, each lattice spacing was converted into the lattice constant $a$ by:

$$a = \frac{d_{hkl}}{\sqrt{h^2 + k^2 + l^2}} \tag{22}$$

to normalize the lattice spacing.

The measured lattice constant $a$ was equivalent to $a_{\varphi\psi}$, which was the lattice constant of the direction $\varphi\psi$, as shown in Figure 12. The lattice constant $a_{\varphi\psi}$ must be converted into the lattice constant $a_1$, which is in the direction of the bending specimen axis. The lattice constant $a_1$ is given by:

$$a_1 = \frac{a_{\varphi\psi} - a_0}{\left(\cos^2(\varphi) - \nu\,\sin^2(\varphi)\right)\,\sin^2(\psi) - \nu\,\cos^2(\psi)}\,, \tag{23}$$

where this equation was derived from the relationship of the strains under the four-point bending of a rectangular bar. In this study, the Poisson's ratio $\nu$ for each lattice plane was determined by the Kröner model [19]. The diffraction elastic constants according to the Kröner model were obtained from the Rigaku site [20].

The lattice constants $a_{\varphi\psi}$ were measured using the DEM with white X-rays, and the lattice constant $a_1$ in the direction of the bending specimen axis was calculated using Equation (23). The regression line

was determined from the relationship between $a_1$ and $\zeta$, which was the distance from the compressive surface of the bending specimen, as shown in Figure 12. As the position of $\zeta = 3$ mm is a neutral axis of the bending specimen, we defined the value of the regression line at $\zeta = 3$ mm as the stress-free lattice constant $a_0$. As a result, we obtained $a_0 = 0.357985$ pm. The bending strain $\varepsilon$ was calculated using the strain-free lattice constant $a_0$.

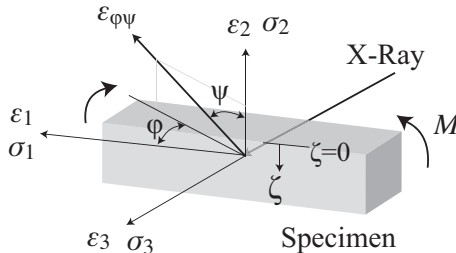

**Figure 12.** Relationship between directions of measured strain and bending specimen axis.

The strains were measured at $\zeta$ from 0.2 to 5.7 mm with a step size of 0.5 mm. The many strains at every $\zeta$ were obtained by the DEM with white X-rays, but these values varied. Therefore, the strain at $\zeta$ is expressed as the mean of the strains. Figure 13 shows the distribution of the bending strain measured by the DEM with white X-rays. The red dotted line in the figure indicates the distribution of the applied strain. The applied strain was measured by the strain gauge that was attached to the tensile surface of the specimen. The measured strains correspond to the applied strains, as shown in Figure 13.

The DEM with white X-rays can be used to measure the strain of coarse-grained materials, although the precision of the DEM with white X-rays is insufficient.

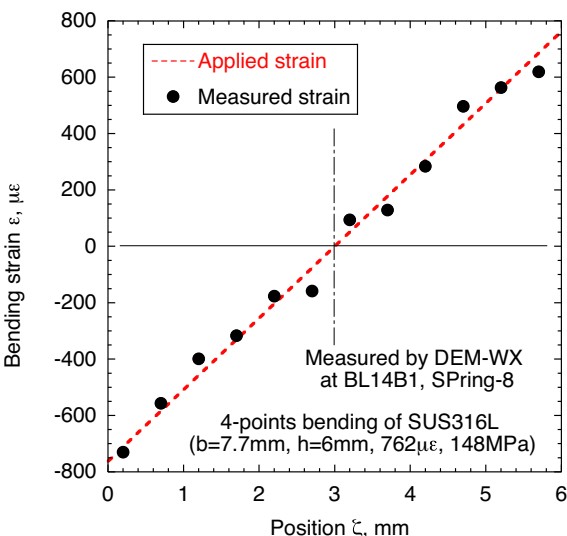

**Figure 13.** Strain distribution measured using DEM with white X-rays. The DEM is the double-exposure method.

## 4. Conclusions

In this paper, we proposed the DEM with white X-rays, and the method for processing the image data with a CdTe pixel detector was discussed. The bending strains were measured using the DEM with white X-rays as a feasibility study. The conclusions obtained in this study are summarized as follows:

(1)     We proposed a new X-ray stress measurement method, a double-exposure method with a CdTe pixel detector, and synchrotron white X-rays. This method is useful for coarse-grained materials and large sample sizes.

(2)     The procedure and system for processing the image data measured with the CdTe pixel detector were discussed. As a result, we found that an X-ray energy image can be obtained from the difference between the energy threshold images.

(3)     The double-exposure method was applied to the diffraction spot image obtained from the above procedure. The distribution of the bending strain measured by our proposed method corresponded to the applied bending strain.

The X-ray stress measurement using the DEM with synchrotron white X-rays would develop by improving precision of resolving X-ray energy of the CdTe pixel detector and progress of data analysis technique in the near future.

**Author Contributions:** Conceptualization, K.S.; Development of Detector and Data Acquisition, H.T. and C.S.; Methodology and Experiments, all authors; Data Analysis, K.S.; Writing—Original Draft Preparation, K.S. All authors read and approved the final manuscript.

**Funding:** A part of this research was funded by the Japan Society for the Promotion of Science, JSPS KAKENHI for Scientific Research (C) 17K06046 and 20K04156.

**Acknowledgments:** The synchrotron radiation experiments in SPring-8 were performed with the approval of QST and JASRI as follows: No. 2018A3653, No. 2018B3653, No. 2018B3684, No. 2019A1636, and No. 2019B3684. The QST experiments were supported by the Nanotechnology Platform of the Japanese Ministry of Education, Culture, Sports, Science and Technology as follows: No. A-18-QS-0032 and No. A-19-QS-0037.

**Conflicts of Interest:** The authors declare that they have no competing interests.

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
