# Peer review of "Double-Exposure Method with Synchrotron White X-ray for Stress Evaluation of Coarse-Grain Materials"

_qubs, doi:10.3390/qubs4030025_

Round 1
Reviewer 1 Report
This paper is devoted to residual stress measurements by x-ray diffraction on engineering materials (i.e. machine elements for instance) where high energy x-rays are mandatory to go through metallic parts few millimeters thick. In addition, the diffracting grain size are three times larger than the x-ray beam used which means a single crystal configuration has to be considered. Diffraction with monochromatic x-rays occurs then for particular directions called pole directions. For such measurements, 2D detectors are mandatory to reduce recording times as well as specific XRD analysis to get residual stresses (see Ortner, Thin Solid Films, 2013 for instance)
High energy x-rays are available at synchrotron radiation facilities and then a lot of techniques have been developed for 20 years to access 3D microstructural information different kind of materials using combination of tomography, XRD and also EXAFS measurements (see Poulsen, Journal of Applied Crystallography, 2012 and related experiments at ESRF Grenoble – France for instance).
Here, the originality is linked to the use of white beam associated with an energy sensitive 2D detector to apply a method called double exposure method – DEM and already developed by the authors (see ref. 8) for monochromatic x-ray beam. The main difficulty seems to be related to the data treatment because of the two detector positions and energy scan and the authors have elaborated a careful procedure to overcome these difficulties. The obtained results are in agreement with what is expected during this conventional deformation experiment. The method seems time consuming, several technical deficiencies appear all along the manuscript, uncertainties are missing there are no references to other methods based on Laue patterns for which specific software have been developed etc.
The paper is written like a technical report which means that the scientific context or interest of such developments for material science and engineering issues is not detailed as well as existing XRD technics by other groups. The authors are known in the “residual stresses” community for developing such measurements at synchrotron facilities and their major contribution is disseminated through conference papers (see references 6 to 8).
To my opinion, this contribution needs to be improved at different levels to meet the standards of regular journals.
Reviewer 2 Report
Stress measurements of coarse-grained material using synchrotron X-ray diffraction are difficult because the diffraction patterns of these materials are spotty. In addition, for the transmitted X-ray beam the center of the diffraction pattern is unknown. Current manuscript describes the double-exposure method with synchrotron white X-Ray for stress evaluation of coarse-grain materials, as well as and the method for processing the image data with a CdTe pixel detector. Based on the importance of the topic the manuscript is acceptable for the publication in QBS in its current form.
Author Response
I really appreciate your review.
Our manuscript was revised according to the other reviewers' comments.
Please confirm the highlighted manuscript.

Reviewer 3 Report
Dear authors.
I was very interested by your work, especially with the part dealing with pixel detector, this paper deserves to be published with just some slight modifications. I am not a crystallographer neither a physicist but here is what I understood from your work:
1) We can estimate the stress on a material by measuring the deformation, both linked by the corresponding tensor.
2) We can calculate the deformation in a material by measuring the variation of the interreticular (interplanar) distance "d".
3) To do so we can use the X-ray diffraction and the Bragg law λ = 2dsin(ɵ).
4) As your sample is big enough, you cannot use the rotating crystal technique and you must use the Laue technique.
5) As your sample is a coarse grain material, the diffraction pattern is spotty, the diffraction center depend of each grain and so the diffraction angle (2 ɵ) cannot be determined. To overcome these difficulties you have proposed a new technique (DEM) in your 2018 paper. This former study was done with a monochromatic X-ray at 30 KeV.
In the paper you submit now, you propose to improve your method by using a CdTe pixel detector and a white X-ray beam, at synchrotron. This allow you to collect diffraction data from 60 KeV to 80 KeV with energy discrimination. Applying your DEM technique to this more complete data set allow you to significantly improve your results.
----------------------------------------------------
I have some requests concerning the part of your paper dealing with the threshold and the energy filtering :
- At the beginning of your 3.1 paragraph you talk about a window-type comparator (window = 2 thresholds) but then you work only with one threshold.
- In the beginning of the 3.2 paragraph, you write " Constructing the threshold X-ray energy from the image data with the threshold voltage is an important aspect of this work, and requires enormous computing times" but you don't explain what it means and how you are processing your images.
So I don't understand how works the electronic of your comparator, and I don't find the response in your 2 previous papers dealing with your detector (2017 and 2019), there is nowhere electronic synoptic diagram for one pixel.
In yours previous paper you say that there is 3 thresholds and one 24 bits counter for each pixel. I guess you can choose to use one threshold with one 24 bits counter, or 2 threshold with two 12 bits counters or three threshold with three 8 bits counters ? Why don't you use two thresholds to get a real window-type comparator ? I read in yours previous paper that your intensity maximum was 24 000 cps/pixel. Then if you use two thresholds with two 12 bits counter your are limited at 4096 counts, may be it is the reason. But even so, you could collect images at a frame rate of 10 frames by seconds and it should be ok. I don't find any information on the frame rate of your detector ?
These considerations, and the fact that you need an enormous computing time to process the images without saying why, drive me to don't understand this important point. Because "normally" you just need to subtract the image i+1 from the image i, pixel by pixel, to get the energy image. One possible explanation could be that rather to have an individual threshold for each pixel you have only one common threshold for your whole ASIC chip (in fact 3 thresholds / 3 DAC). If so I understand things but you have to explain clearly this in your paper. If not please explain how it works and how you justify your choices. I think you also should give some information on your DAC : number of bits and value of the LSB in mv or in KeV (that means the energy window in which you can tune your threshold and the minimum step of tuning) because energy filtering is an essential point in your work.
At the beginning of paragraph 3.1 you explain how works the pixel comparator. I think you should insist on the fact that the photon pulse and the threshold are both negative voltage, and so things are reversed, because for a non specialist it is not so easy to understand, or better you can give a little diagram (instead of 4 b,c) to show some photons pulses with different energy and the threshold level varying inside two limits ( your "window-type" ?).
At the end of the introduction you write : " The DEM is a creative technique, but the DEM is not suitable for high-energy X-rays as it uses the PILATUS detector". You have to suppress this sentence or modify it "as it uses our PILATUS detector" or "as it uses Si sensor pixel detector". I think your PILATUS is a silicon sensor pixel detector because you use it at 30 KeV in your previous study, but Dectris also commercialize big CdTe pixel detector. I don't think Dectris let you access "inside the detector" to do all the energy calibrations that you need but Dectris should be able to do it for you, it is just a question of "commercial negotiation" with them I guess.
There is also an XPAD CdTe big detector that exist and you can access to all the tuning inside the detector you need to do all the calibrations you want. Although XPAD is a one threshold detector there is also a special window energy mode, clustering pixels, and calibrated them with different energy threshold (under patent licence, personal communication from Pierre Delpierre, inventor of pixel detector). Note that the society ImXPAD does not exist anymore, theses detectors are now provided by CEGITEC company (France) under the name of RebirX detector.
There is also a little GaAs pixel detector, Lambda detector from X-spectrum company in Hamburg Germany. The GaAs sensor can detect X-ray in the range 60 - 80 KeV with less efficiency than CdTe but with more stability and may be with such a detector you should not have the difference in the threshold before and after measurement as shown in fig 7a.
Kind Regards.
Author Response
I really appreciate your kind comments. Your advice on data acquisition help our further study.
Please confirm the highlight manuscript and the answer.

Round 2
Reviewer 1 Report
No comments
Author Response
Dear Reviewer1,
Thank you for your reviewing.
My manuscript was revised. The highlighted file (pdf) as reference was also prepared. Please confirm it.
Best regards,
K. Suzuki
